# Implementation of Remote Control for the AM 524 Antenna Amplifier Unit System in SAC Chambers

**Leszek Nowosielski** , **Jan M. Kelner**, **Bartosz Dudziński ***  **and Mateusz Rychlicki**

Faculty of Electronics, Military University of Technology, 00-908 Warszawa, Poland
* Correspondence: bartosz.dudzinski@wat.edu.pl

**Abstract:** In the rapidly evolving landscape of modern technology, particularly in telecommunications and pervasive computerization across diverse sectors, the value of information has soared, becoming the linchpin of success in politics and business alike. With the majority of information now flowing through various computing devices, safeguarding them from unauthorized interception has assumed paramount importance. A critical threat in this context emanates from unintentional electromagnetic emissions generated by these devices. Under favorable conditions, these emissions can be exploited by unauthorized entities to reconstruct processed information, a phenomenon known as electromagnetic infiltration. Such emissions, correlated with useful information and conducive to its reconstruction, are termed revealing emissions, with the enabling process labelled as electromagnetic information leakage. This article presents the design and construction of a remote control system for managing antenna amplifier blocks within the AM 524 antenna system, dedicated to investigating information leakage from multimedia devices. The system facilitates remote switching of the five inputs on antenna amplifiers GX 525, GX 526, and GX 527 from a PC, utilizing specialized software. The authors provide an overview of the AM 524 antenna system, elucidate the design concept behind the remote control system, and highlight the central component—the ADAM 6052 module. Additionally, the article introduces the controlling software. It encompasses the device's construction, including component details, connection schematics, and images of the assembled system, along with a verification process confirming its operational accuracy. Furthermore, the article outlines the application of the proposed solution in assessing the effectiveness of shielding within SAC chambers, employing the measurement methodology specified in accordance with the EN 50147-1:1996 standard. This additional information underscores the practical utility and relevance of the presented remote control system in the context of electromagnetic shielding evaluation for secure environments. Additionally, to assess the effectiveness of the proposed commutator solution, measurements were conducted to evaluate the shielding efficiency of the SAC chamber using a modified coaxial cable. The results of the shielding efficiency of the SAC chamber measurements for the proposed and classical solutions are also presented.

**Keywords:** EMC; antenna array; remote control; electromagnetic infiltration



## 1. Introduction

One of the most common research areas in modern electromagnetic compatibility laboratories involves conducting a set of tests aimed at measuring the emissions of devices powered by electrical energy. These emission tests encompass measuring both radiated and conducted disturbances levels [1–4]. Such investigations typically require specialized equipment, including:

- Measurement sensors: antennas, artificial networks, absorption clamps;
- High-frequency signal level meters: measurement receivers, spectrum analyzers;
- RF signal generators;
- RF switches;

- Computers with measurement control software.

In most electromagnetic compatibility laboratories, the aforementioned equipment is installed in two adjacent rooms, separated by an electromagnetically tight barrier. One of these rooms is an SAC chamber, primarily designed to isolate the tested device from external electromagnetic environments. This isolation is achieved through an electromagnetic barrier, an integral part of the SAC chamber, which comprises a highly shielding enclosure (with approximately 100 dB shielding effectiveness), signal, and power filters. By isolating the tested device from the external electromagnetic environment during emission measurements, it becomes possible to distinguish the emitted disturbances from the device under test, which is the subject of the measurements, from external environmental interferences. For this purpose, only the tested device and measurement sensors such as antennas, artificial networks, and absorptive clamps are placed within the SAC chambers. Other equipment (e.g., measurement receivers, RF switches) required for conducted or radiated emission measurements, which might be potential sources of additional disturbances, are situated in an adjacent control room.

In order to expedite measurements and properly configure the emission level testing setup, it is necessary to connect the output of the respective measurement sensor to the input of the high-frequency signal level meter. For radiated emission measurements, measurement antennas covering various frequency ranges are connected to the input of the signal level meter. In practice, switching the antenna modules, such as the active antenna system R&S®AM524, is achieved using a controlling unit, R&S®GS525, through fiber-optic cables R&S®GS525K1 (Figure 1).

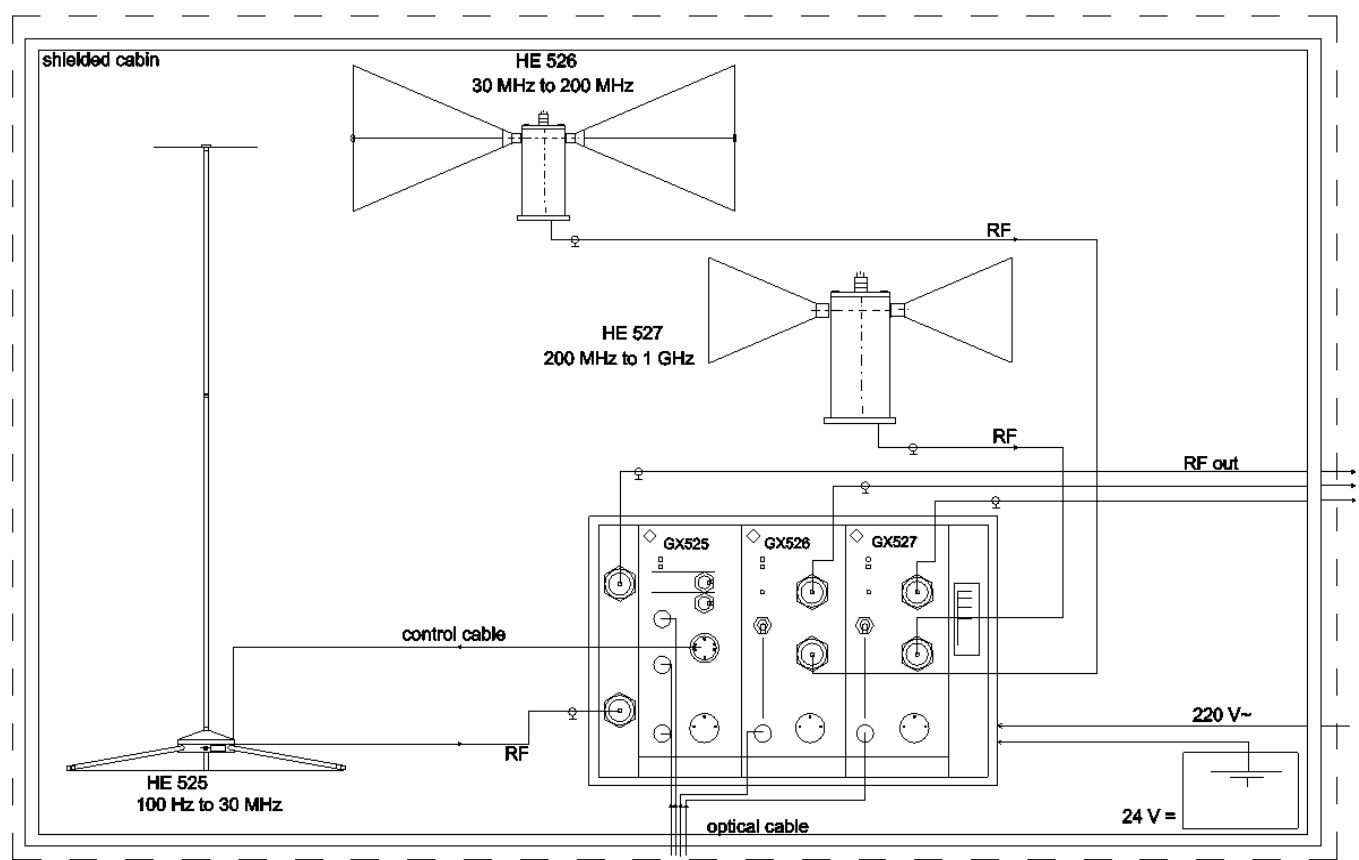

**Figure 1.** Configuration of the active AM 524 antenna system.

The low-noise active antenna system R&S®AM524 has been designed for measuring low-level signals in SAC chambers. The operating frequency range of the antennas spans from 100 Hz to 1000 MHz and is divided into three subranges:

- From 100 Hz to 30 MHz—rod antenna HE 525;
- From 30 MHz to 200 MHz—dipole antenna HE 526;
- From 200 MHz to 1000 MHz—dipole antenna HE 527.

The GX 525, GX 526, and GX 527 connector units (Figure 1) within the AM 524 antenna system provide power to the antennas and facilitate control of the antenna system.

Example block diagrams of two configurations for measuring radiated and conducted disturbance levels using the OSP 130 switch installed inside and outside the SAC chamber are shown in Figure 2 and Figure 3, respectively.

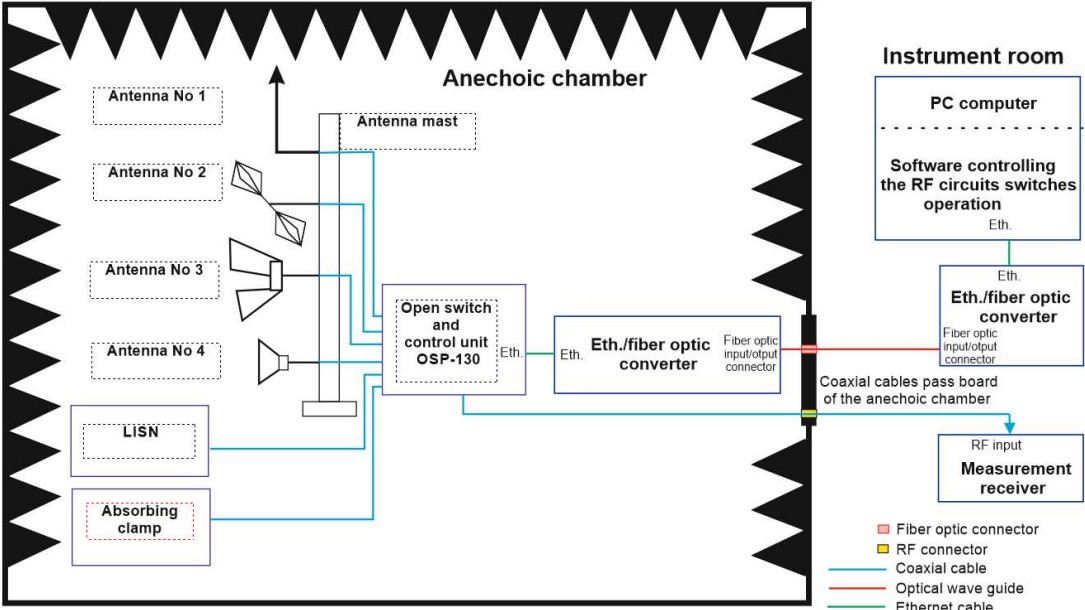

**Figure 2.** Block diagram of the circuit for disturbance emission level measurement with the OSP-130 commutator installed inside the SAC chamber.

The measurement setup configuration with the OSP 130 switch installed inside the SAC chamber, as depicted in Figure 3, has a notable drawback: an increased level of disturbances generated within the SAC chamber. This increase is due to the placement of the OSP 130 switch inside the chamber. The unintentional radiated and conducted disturbance emissions produced by the switch can interfere with the measurement results of the emitted disturbances from the device under test. This device is situated in the same electromagnetic environment as the OSP 130 switch, within the SAC chamber. Another drawback of this setup is the requirement for fiber-optic converters in both the SAC chamber and the control room. These converters enable the broadband transmission of remote control commands to the OSP 130 switch via the LAN/USB interface.

A better solution, in terms of the level of disturbances generated within the SAC chamber, is presented in the measurement setup configuration shown in Figure 3. In this configuration, the OSP 130 switch is installed outside the SAC chamber, in the control room. However, this solution is not without its drawbacks, as it requires routing a significant number of coaxial cables through the SAC chamber wall, and connecting the outputs of the B 102 module in the OSP 130 switch to the inputs of individual measurement sensors. Routing numerous coaxial cables through the SAC chamber requires creating an equivalent number of feedthroughs in the chamber enclosure. This can potentially compromise the shielding effectiveness of the chamber where these feedthroughs are installed.

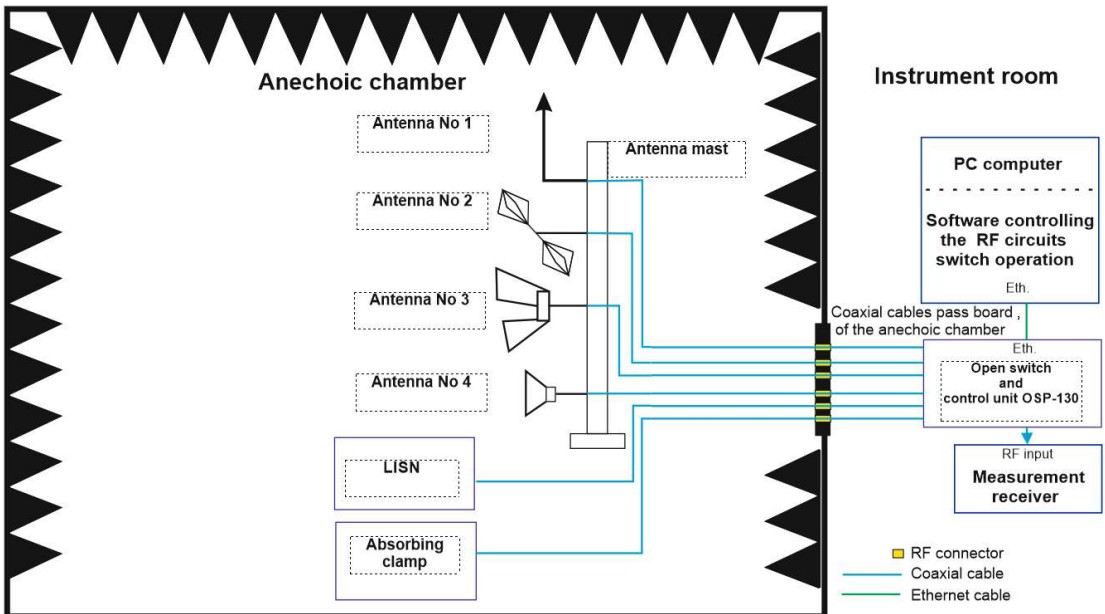

**Figure 3.** Block diagram of the circuit for disturbance emission level measurement with the OSP 130 commutator installed outside of the SAC chamber, in the instrument cabin.

In the subsequent part of this article, the authors present an innovative method used in the Electromagnetic Compatibility Laboratory of the Military University of Technology at the Faculty of Electronics for control communication to the controlling unit R&S®GS525, responsible for switching the antennas within the R&S®AM524 antenna system, which makes use of an LED installed along with a 10 Ohm carbon resistor in a coaxial cable. That solution within the SAC chamber lacks the drawbacks associated with the configurations depicted in Figures 2 and 3. These prior configurations are based on conventional OSP 130 switch solutions, in which the executive circuits (RF switches) and control circuits, including the software controlling them, are integrated within a single switch enclosure.

Based on the analysis of the potential configuration variants for measuring radiated and conducted disturbance levels using the OSP 130 switch, it can be concluded that the new method should enable:

- Switching RF circuits, equal in number to the quantity of measurement sensors installed inside the SAC chamber, using executive circuits of the OSP 130 switch located outside the SAC chamber;
- Remote control of the executive circuits switch using measurement process control software;
- Switching the individual measurement antenna outputs to the input of the RF signal level meter via coaxial cables;
- Minimizing the shielding effectiveness reduction of the SAC chamber at the location of feedthroughs connected to individual measurement sensor outputs and the input of the RF signal level meter.

## 2. Concept of Remote Control for the Operation of Antenna Amplifier Blocks in the AM 524 System

In the following section of this article, the authors describe the concept of remote controlling the operation of the GX 525, GX 526, and GX 527 antenna amplifier blocks from a PC (Figure 4), utilizing the ADAM-6052 I/O control module for this purpose. The ADAM module was configured using the "ADAM-5000TCP-6000 Utility" software provided by the manufacturer, as well as the "Switcher_No_3 v_3" software, which is responsible for toggling the appropriate inputs on the respective amplifiers.

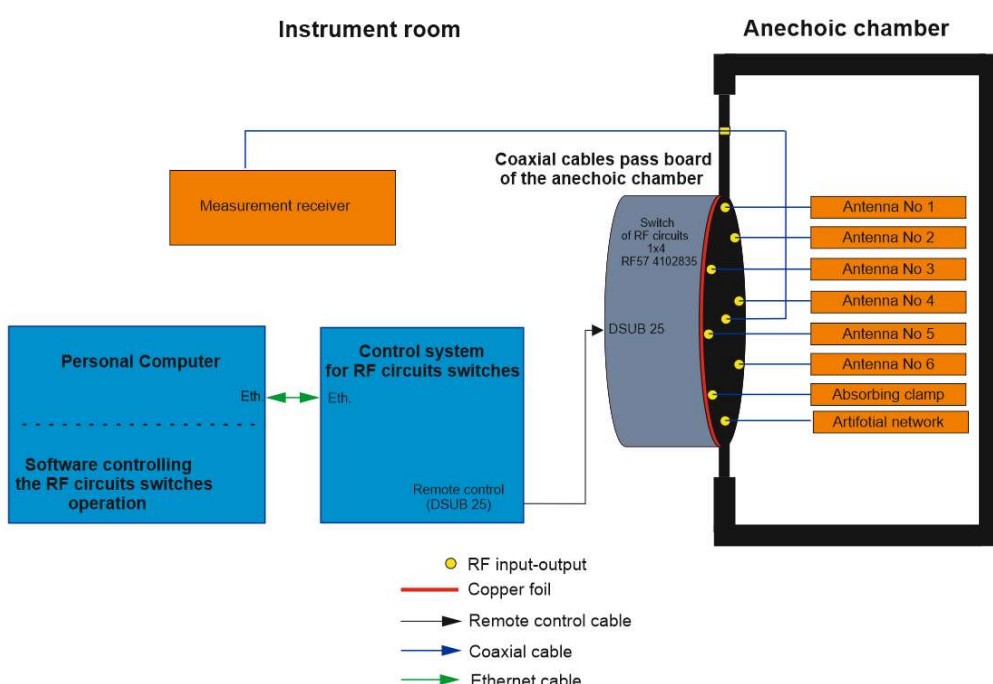

**Figure 4.** Simplified block diagram of the proposed solution for the HF circuits commutator.

The concept of the remote control system for the antenna amplifiers, which includes a suitably modified coaxial cable and a PC connected to the designed setup via an Ethernet network cable, is presented in Figure 5.

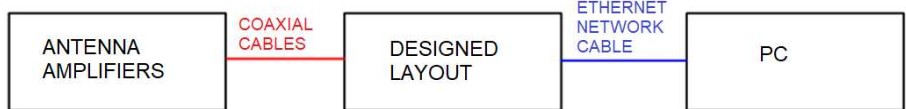

**Figure 5.** Concept of remote control system for antenna amplifiers.

In Figure 6, a method of installing an LED in a coaxial cable is illustrated. The cathode of the LED is connected to the transmission wire, while the anode of the LED, along with a 10 ohm carbon resistor soldered in series, is connected to the shielding braid with two single-core soldered wires. The carbon resistor restricts the current to 58 mA, effectively limiting the heat generated at the soldered connection, thereby resolving the issue of solder joint deterioration caused by excessive solder temperature. The entirety of the modified cable is placed within a male SMA connector. The transmission wire's purpose is to deliver a 3.3 V supply voltage to the LED. The shielding braid serves as the ground, resulting in the coaxial cable forming a closed circuit with the LED.

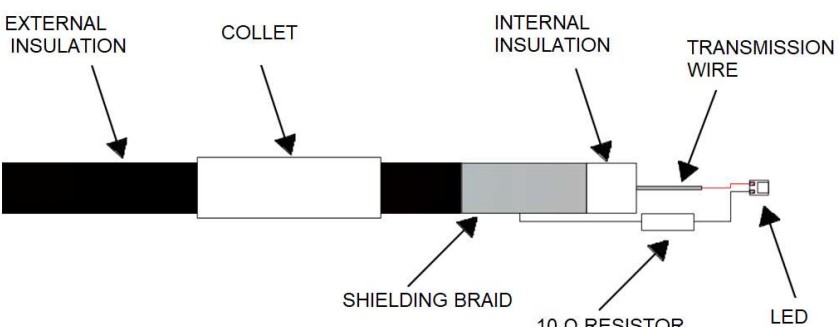

**Figure 6.** Modified coaxial cable.

Figure 7 presents the arrangement of individual components within the designed system.

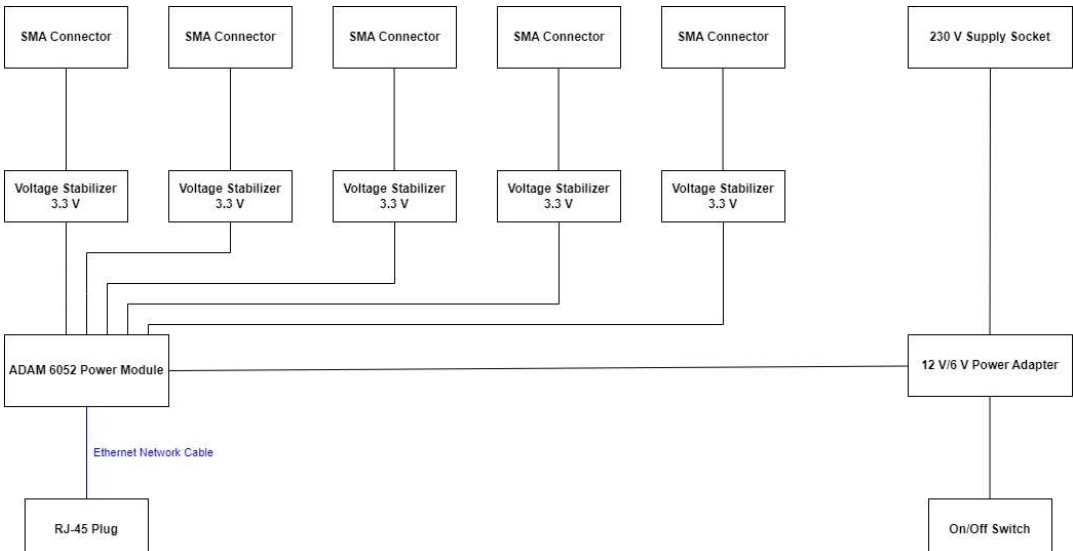

**Figure 7.** Arrangement of individual components within the designed system.

The exterior view of the ADAM 6052 Control Module is shown in Figure 8. This module enables the integration of automation systems using internet technology. It serves as a central component of the device, facilitating the conversion of remote control commands intended for amplifier unit manipulation. These commands are transmitted from the PC with control software to the ADAM 6052 module via the TCP/IP protocol. The module converts these commands into 0 V/12 V direct voltage signals, which are then delivered to the digital outputs of the ADAM 6052 module. In the presented design, five outputs from the ADAM 6052 module were utilized. The signal from these outputs was then directed to voltage stabilization circuits, which is essential for ensuring the proper operating conditions of the LED. The diode is placed in one of the SMA connectors of the coaxial cable, which is connected to the antenna amplifier on one end, and to the SMA connector within the described device (henceforth referred to as the "control component") on the other end. All device components were housed within a plastic enclosure.

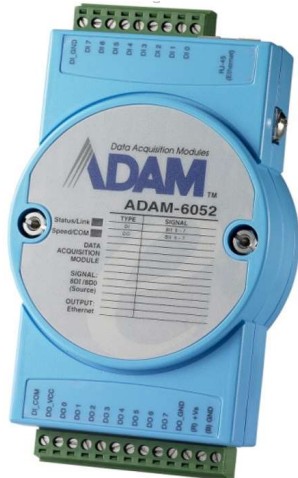

**Figure 8.** External view of the ADAM 6052 control module.

The LED serves as a source of white light. When the diode emits a pulse of light, the antenna amplifier system reacts, transitioning from an inactive to an active state.

The utilization of the coaxial cable and an LED allows for a cost-effective alternative to expensive dedicated fiber-optic cables. Additionally, the coaxial cable meets electromagnetic shielding requirements and boasts high resistance to physical damage.

## 3. Control Software for the Operation of Antenna Amplifier Blocks in the AM 524 System

The control software developed for the operation of the GX 525, GX 526, and GX 527 antenna amplifier blocks is a critical component of the AM 524 system. This software enables the remote manipulation and switching of antenna inputs from a PC interface. The software is designed to work seamlessly with the ADAM-6052 control module, allowing users to efficiently manage the functionality of the antenna amplifiers.

Key features of the control software include:

- Remote Control Interface: The software provides a user-friendly interface on the PC, allowing users to interact with and control the amplifier blocks remotely;
- Antenna Input Selection: Users can select and switch between the various antenna inputs on the GX 525, GX 526, and GX 527 amplifier blocks using the software interface;
- TCP/IP Communication: The software establishes communication with the ADAM-6052 control module through the TCP/IP protocol, facilitating a seamless data exchange between the PC and the control module;
- Real-time Feedback: The software offers real-time feedback on the status of the amplifier blocks, indicating which inputs are active, and allowing users to monitor changes as they occur;
- Error Handling: The software includes error detection and handling mechanisms, ensuring reliable operation and notifying users of any issues that may arise;
- Centralized Management: Multiple instances of the software can be managed centrally, providing the ability to control multiple AM 524 systems simultaneously;
- Customization: The software may offer customization options, such as user-defined presets for antenna input configurations, to enhance user convenience.

Using the control software's intuitive interface and its integration with the ADAM-6052 module, operators can effectively manage the antenna amplifier blocks within the AM 524, enhancing both the performance and reliability of the system. To ensure the proper operation of the antenna amplifiers' remote control system, the ADAM 6052 device and the dedicated control software for switching the system's outputs need to be appropriately configured. To configure the remote module controllers, the authors employed the "ADAM-5000TCP-6000 Utility" software provided by the manufacturer. For system output switching, they used the "Komutator_Nr_3" application running on Windows 10.

The software configuration process began by establishing a connection between the master unit (PC) and the designed device's RJ-45 Ethernet port using a LAN cable, enabling communication with the ADAM 6052 module.

The initial steps involved adjusting the network settings of the PC's network adapter. Manual changes were made to the IP addresses and DNS settings to ensure that the remote control module and the master unit operate within the same network. Example settings are shown in Figure 9.

Subsequently, the "ADAM-5000TCP-6000 Utility" program was launched. The ADAM 6052 devices were automatically detected, with the identified modules and their assigned IP addresses displayed on the left side of the user interface. The "PING" button was used to verify that the module is correctly connected the computer, as shown in Figure 10.

After selecting a specific module by clicking its IP address with the left mouse button, a series of different tabs appear in the program window for making adjustments. Within the "Network" tab, users can modify default network settings such as the IP address, subnet mask, default gateway, and hostname. To properly configure the device, users need to set the network parameters in which the control system will operate. Any changes made must be confirmed by clicking the "Apply" button and entering a password (the default password is "00000000") see Figures 11 and 12.

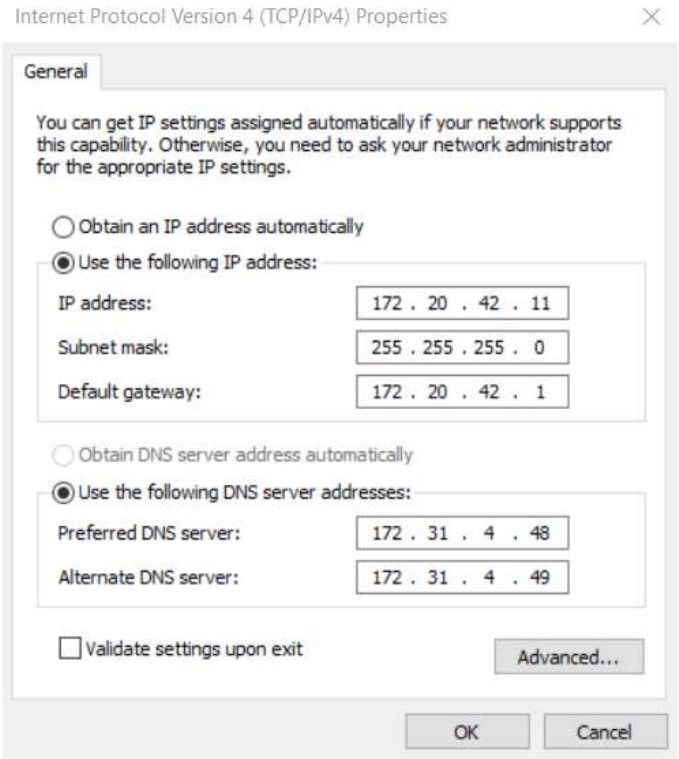

**Figure 9.** Network adapter settings on the computer.

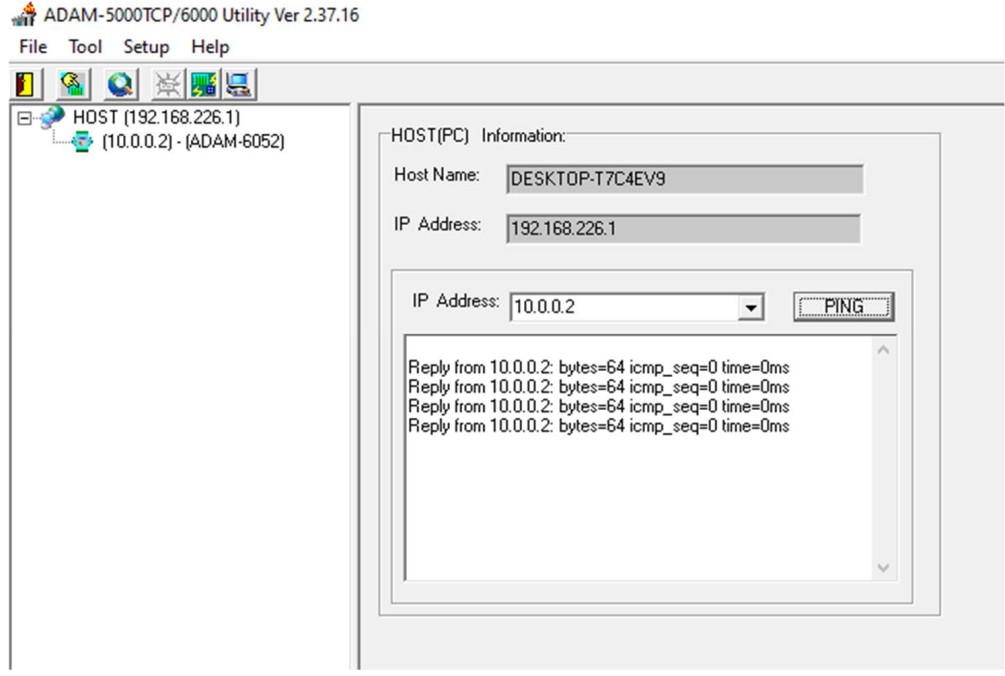

**Figure 10.** Main screen of the "ADAM-5000TCP-6000 Utility" program.

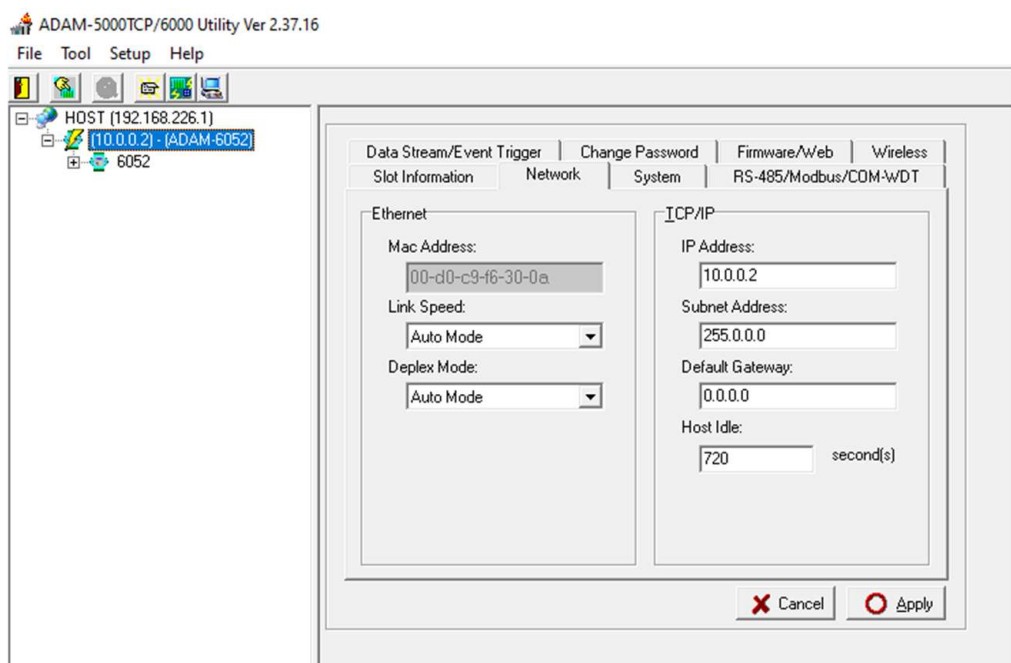

**Figure 11.** "Network" tab with network settings shown.

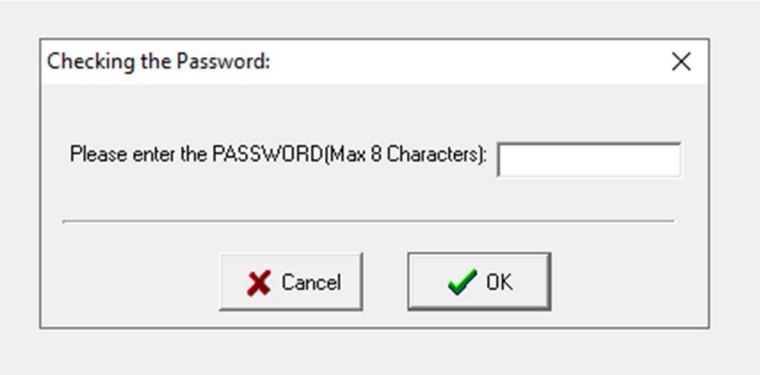

**Figure 12.** Access password dialog box.

The control of switching individual antenna amplifiers is carried out using the "Komutator_Nr_3v_3" program. The user interface of this software is shown in Figures 13 and 14.

The buttons labelled "Input 1" to "Input 5" are used to select the antenna amplifier that is to be remotely activated. Clicking one of the five buttons corresponding to an amplifier's name will turn it on. It should be noted that only one input can be active at any given moment. The "Disable inputs" button allows users to disconnect power from all outputs of the remote control system simultaneously.

In the software's settings menu, users can customize the button labels to match specific devices. They can also set the initial position of the graphical user interface upon program startup by specifying the X and Y coordinates. The IP address of the ADAM 6052 module is also displayed in this menu, and any changes to this address can be made in the "config.txt" file, as shown in Figure 15.

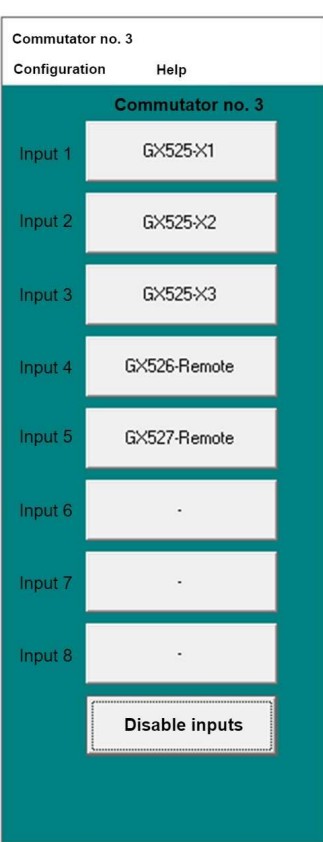

**Figure 13.** User interface of the "Komutator_Nr_3" program.

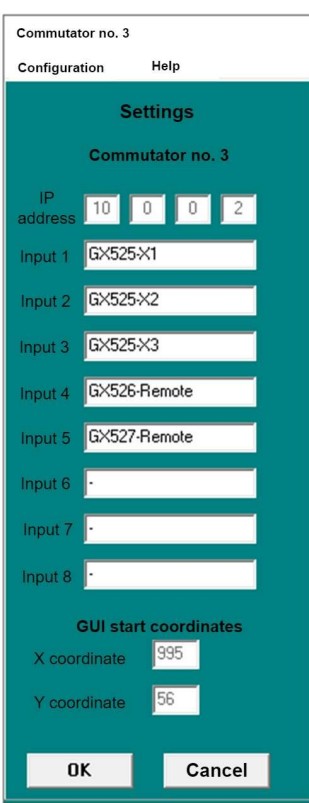

**Figure 14.** Settings of the "Komutator_Nr_3" program.

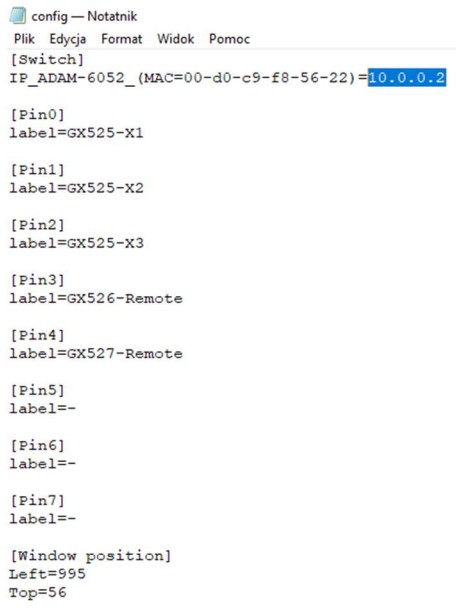

**Figure 15.** Contents of the "config.txt" file.

Figure 15 depicts the "config.txt" text file. Within the highlighted area (blue rectangle), users should input the IP address of the ADAM 6052 module with which the "Komutator_Nr_3" software is intended to work with.

## 4. Development of Remote Control System for Antenna Amplifier Operation

The designed remote control system for the operation of the AM 524 antenna amplifier block comprises two groups of components that together form a functional unit. The first group of components, forming the control part, includes:

- ADAM 6052 control module (1 pc);
- 12 V/6 A DC power supply (1 pc);
- 3.3 V voltage regulators (5 pcs);
- Z39W KRADEX enclosure (1 pc).

These components, housed within the enclosure, form the control section responsible for generating control signals in response to commands transmitted from the PC:

- Female SMA connectors (5 pcs);
- Male SMA connectors (10 pcs);
- 6 A 240 V AC power socket with filter (1 pc);
- CE power socket with 20 A fuse (1 pc);
- RJ-45 panel connector (1 pc);
- Power switch (ON/OFF) (1 pc);
- Through-hole indicator LEDs (green) (5 pcs);
- Ethernet cables (2 pcs);
- high-power LEDs (white) (5 pcs);
- H155 coaxial cable (5 m),
- 300 Ohm carbon resistors (5 pcs);
- 10 Ohm carbon resistors (5 pcs).

These components were used to create the modified coaxial cable, enabling the transmission of signals from the control section to the inputs of the antenna amplifiers. The arrangement of individual components inside the device is depicted in Figure 16.

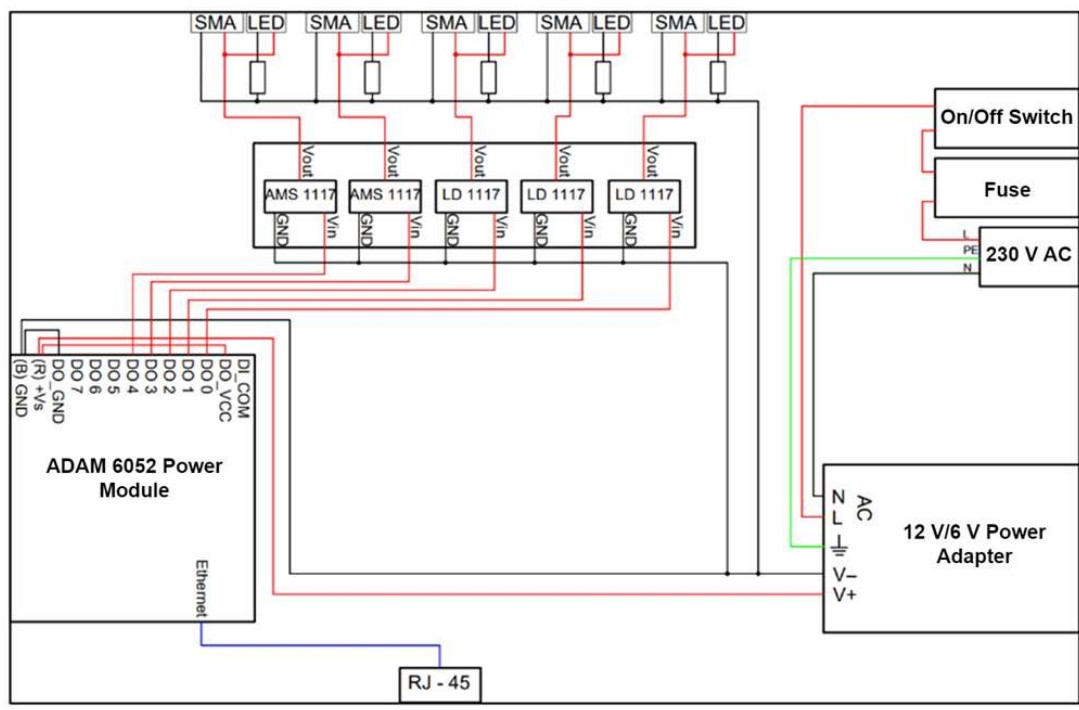

**Figure 16.** Internal connections within the system.

Red lines indicate the power supply and signal wires, while black lines represent the ground wires. The protective wire is green, and the blue line signifies the Ethernet network cable.

The designed system has been housed within a plastic enclosure. In Figure 17, the control part of the system is visible.

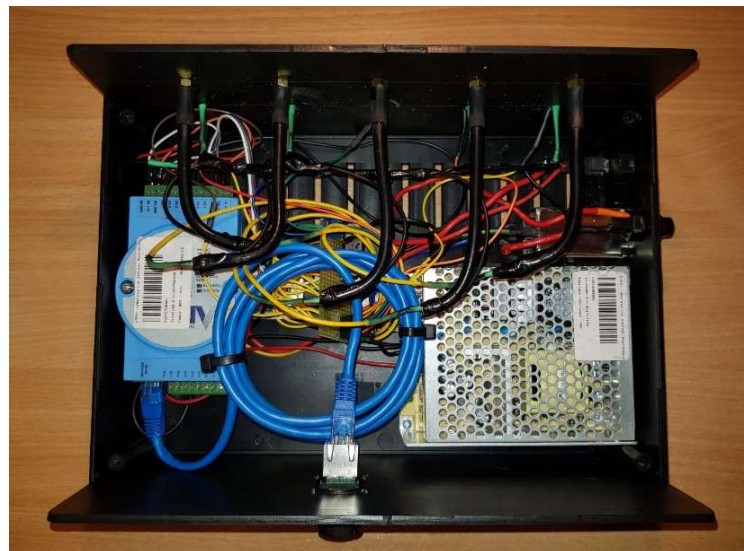

**Figure 17.** Top view of the device (control part) with enclosure section removed.

The rear panel of the device, shown in Figure 18, houses the RJ-45 socket. The antenna amplifiers' remote control system connects to a PC using this connector and an Ethernet cable.

The modified coaxial cable is displayed shown in Figure 18. When correctly connected to the control part, this cable emits a white light source.

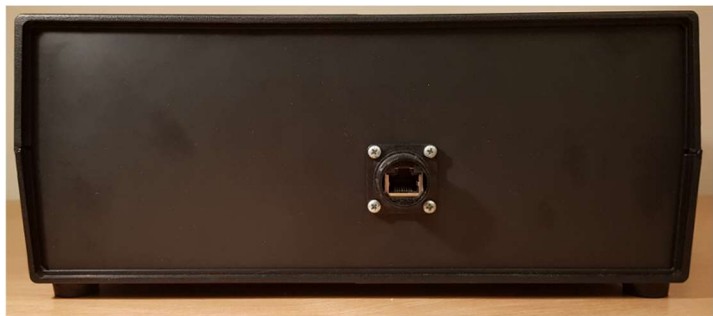

**Figure 18.** Rear panel of the device.

Testing the functionality of the modified coaxial cable is depicted in Figure 19.

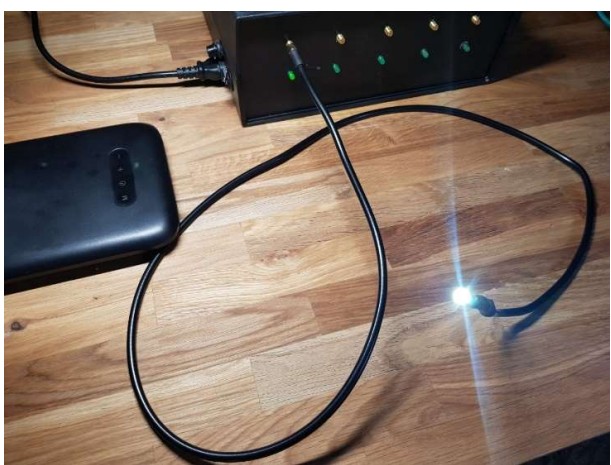

**Figure 19.** Testing the functionality of the modified coaxial cable.

## 5. Testing the System with Antenna Amplifiers

To test the remote switching capabilities of the antenna amplifiers, a test setup was assembled as shown in Figure 20. This setup utilized the schematic of the testing station for verifying the operation of the remote control system for the antenna amplifier block, presented in this article and depicted in Figure 21.

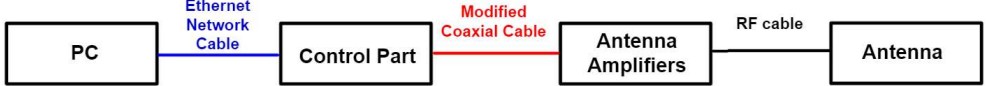

**Figure 20.** Schematic of the setup for testing the remote control system for antenna amplifier operation.

In this testing setup:

- The remote control unit was connected to the computer via an Ethernet cable;
- The antenna amplifiers were connected to the remote control unit using the modified coaxial cables;
- The power supply and stabilizers ensured proper power distribution to the amplifiers;
- The control software, "Komutator_Nr_3v_3", was installed and configured on the computer;
- The purpose of this test setup was to demonstrate the practical functionality of the remote control system in a controlled environment. By following the testing procedure described in the previous section, the operation of the antenna amplifiers could be verified and validated.

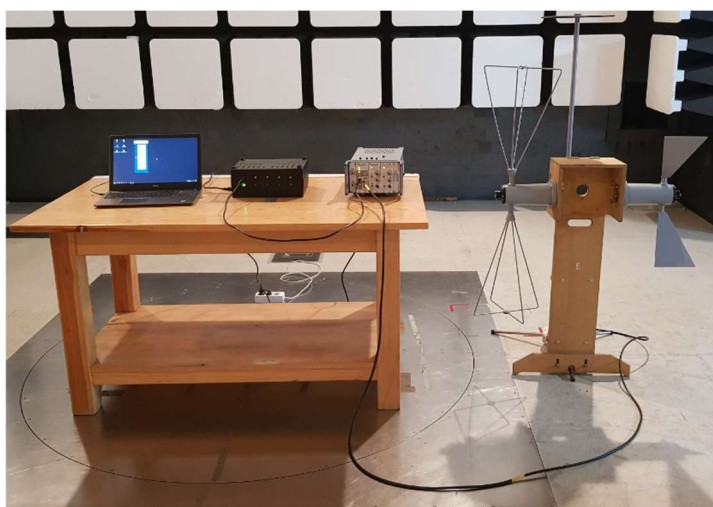

**Figure 21.** Assembled testing setup for evaluating the performance of the remote control system.

Figure 22 showcases the actual physical arrangement of the test setup, while Figure 23 illustrates the schematic representation of the testing station, highlighting the interconnections between the various components. This comprehensive testing process confirmed the efficacy and reliability of the designed remote control system for the antenna amplifier block of the AM 524 system.

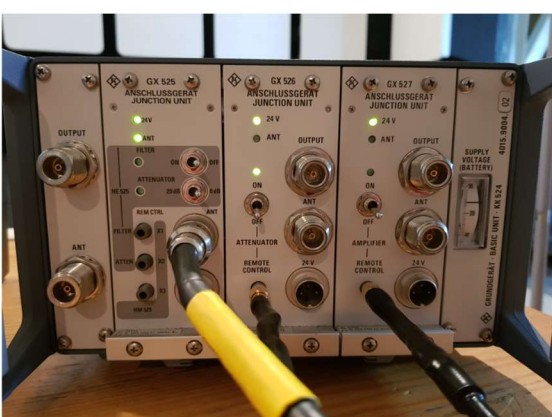

**Figure 22.** Illumination of the indicator LED on the front panel of the GX 526 amplifier unit.

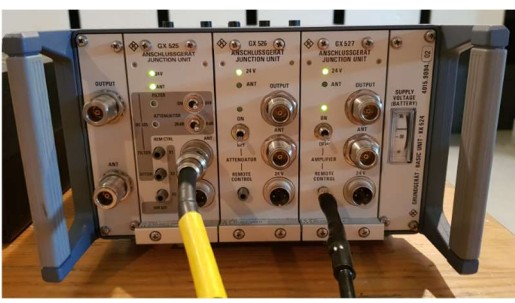

**Figure 23.** Illumination of the indicator LED on the GX 527 amplifier unit.

The correct operation of the system is indicated by the illumination of the corresponding signaling LED when a specific input is activated on the front panel of the main unit housing the antenna amplifiers.

In Figure 22, the front panel of the main unit is displayed, featuring a glowing green LED located next to the attenuator switch of the GX 526 amplifier. The illumination of the LED, observed after connecting and configuring the remote control system and activating

the "GX 526—Remote" output from the PC while the switch is in the "off" position, confirms that the designed system functions correctly.

Figure 23 below shows the verification of the remote control system's operation for the remaining antenna amplifiers.

The aim of this article is to introduce a solution for an RF circuit switch commutator designed for use in SAC chambers. This solution is distinguished by the absence of shortcomings found in the configurations depicted in Figures 2 and 3. Traditional solutions typically integrate the actuating circuits (RF switches) and the control circuits for the actuating systems within a single enclosure. In contrast, the proposed RF circuit switch commutator separates the actuating circuits, built upon RF circuit switches, from the remote control system governing the commutator's operation. This design ensures minimized shielding efficiency losses within the SAC chamber, particularly at the installation points of coaxial cable penetrations connected to individual measurement sensor outputs. This is achieved by employing a modified coaxial cable.

To evaluate the effectiveness of this proposed commutator, the authors conducted measurements assessing the shielding efficiency of the SAC chamber using the modified coaxial cable. Figure 24 presents these results as frequency domain plots differentiated by colors. The obtained shielding efficiency values of the proposed solution (red plot) are juxtaposed against the efficiency of the same SAC chamber with a classical RF circuit switch commutator solution (blue plot), represented by the OSP 130 device configuration shown in Figure 3. A configuration without an RF circuit switch commutator (black plot) was included as well. These measurements followed the methodology described in references [5–21]. Based on the collected shielding efficiency measurement results, it can be concluded that the proposed RF circuit switch commutator solution successfully minimizes the shielding efficiency losses within the SAC chamber at the installation site of coaxial cable penetrations connected to the individual measurement sensor outputs.

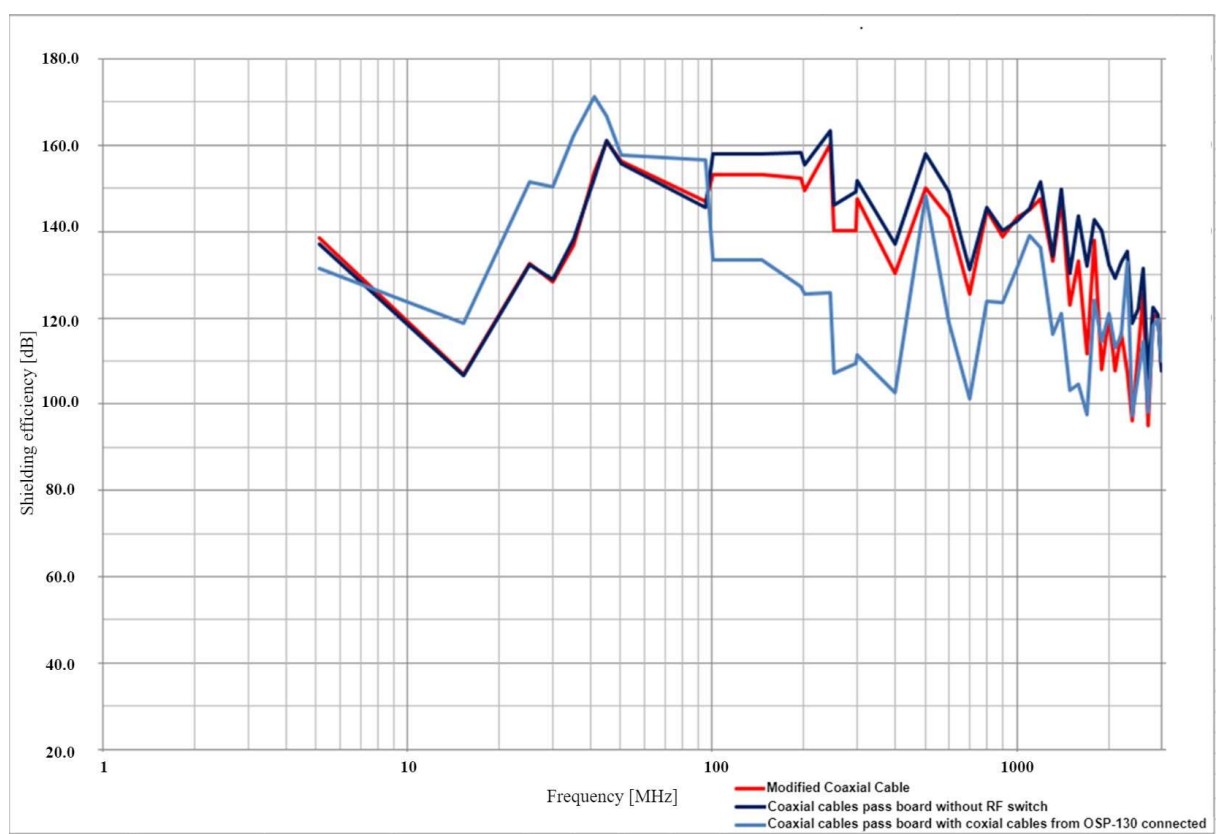

**Figure 24.** Electric field shielding efficiency of the SAC chamber in the installation place of the cable introducing board.

The methodology for measuring the shielding efficiency of the SAC chamber using a modified coaxial cable was derived from EN 50147-1:1996, which provides detailed guidelines for such measurements.

## 6. Summary

In summary, the developed remote control system for the antenna amplifier block of the AM 524 system effectively serves its intended purpose, enabling the remote control of amplifiers without the need for dedicated fiber-optic cables. The choice of using a coaxial cable and SMA connectors required the integration of compact diodes, which would be small enough to fit within the connectors while maintaining a connection to the antenna amplifiers. Consequently, 3.3 V diodes were chosen due to their smaller size compared to their 12 V counterparts. The selection of a coaxial cable was informed by the electromagnetic compatibility needs of the chamber housing the AM 524 system's antenna amplifiers. This choice not only meets these requirements but also mitigates the spread of electromagnetic disturbances outside the cable. Additionally, the inclusion of SMA-SMA adapters on the linear breakout board allows the control segment with the ADAM 6052 module to be positioned outside the SAC chamber.

During the construction of the system, an issue arose with an unstable soldered connection between the diode and the power supply lead, stemming from significant heat generated by a relatively high current. The compact size of the diode compounded this issue, impending heat dissipation, and eventually weakening the soldered connection. To address this, the authors implemented a straightforward solution, which restricts the current flowing through the diode. By incorporating a 10 ohm carbon resistor, the current was effectively reduced to 58 mA. However, this solution resulted in a diminished brightness of the LED.

The verification of the device's functionality, as demonstrated in the article, confirmed that the remote control system was designed and implemented correctly. Additionally, the article presents the outcomes of efficiency measurements for the proposed RF switch circuit solution, accompanied by a comparative analysis of the shielding efficiency between the proposed design and conventional RF switch circuitry.

The proposed RF switch circuit solution enables:

- Switching of high-frequency circuits in correspondence with the number of measurement sensors within the SAC chamber, utilizing external switching units;
- Remote control of these switching units via measurement process control software;
- Routing of individual measurement sensor outputs to the high-frequency signal level meter input using coaxial cables;
- Minimization of shielding efficiency losses within the SAC chamber by employing coaxial cable pass-throughs connected to the outputs of individual measurement sensors.

**Author Contributions:** Conceptualization, B.D. and L.N.; methodology, B.D.; software, B.D.; validation, M.R. and L.N.; formal analysis, L.N.; investigation, L.N.; resources, B.D.; data curation, M.R.; writing—original draft preparation, B.D.; writing—review and editing, L.N.; visualization, J.M.K.; supervision, J.M.K.; project administration, J.M.K.; funding acquisition, J.M.K. All authors have read and agreed to the published version of the manuscript.

**Funding:** This work was financed by the Military University of Technology under Research Project no. UGB/22-863/2023/WAT on "Modern technologies of wireless communication and emitter localization in various system applications".

**Data Availability Statement:** Not applicable.

**Conflicts of Interest:** The authors declare no conflict of interest.

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
