# Peer review of "Implementation of Remote Control for the AM 524 Antenna Amplifier Unit System in SAC Chambers"

_electronics, doi:10.3390/electronics12214416_

Round 1

Reviewer 1 Report

1. There is quantitative performance reported in the abstract.

2. Abstract should only in one paragraph.

3. "For the measurement of revealing emissions, Rohde & Schwarz has developed 21 the R&S®AM524 antenna system." in the abstract is not relevant to your contribution. A very poorly written abstract. 

4. It was mentioned five inputs, but only three amplifiers, please clarify.

5. What are the challenges of the remote control? The motivation of the proposed method is not clear.  Or the authors aim to present a measurement setup only.

6. Figure 6 is not complete.

7. Many of the contents are basically operational, and hard to find scientific contributions, the authors need to show the contribution clearly, e.g. what significant of Figure 8? A commonly seen network setting. Why need to show Figure 11, access password? Similar to the rest of the contents.  

8. For Figure 23, is it far field? What do you mean by shielding efficiency? more than 120 dB is good, but it is known for the coaxial cables, not much meaning to report the modified coaxial cables. 

9. There is no comparison with the existing works.

minor.

Author Response

Thank you for your thoughtful review of our article. We appreciate your feedback and suggestions for improvement. We have carefully considered your comments and want ot answer for yours question

  1. There is quantitative performance reported in the abstract.

In assessing the effectiveness of a Shielded Anechoic Chamber (SAC), a measurement methodology compliant with EN 50147-1:1996 standard was employed. In the context of the conducted measurements, the use of this standard enables an objective and standardized evaluation of the SAC's shielding capability. The measurement methodology described in EN 50147-1:1996 ensures precise and reliable results that can be compared to industry norms and international standards concerning Electromagnetic Compatibility (EMC). The application of this measurement methodology allows for a precise determination of how effectively the SAC protects measurement devices from electromagnetic interference in their environment. This is crucial, particularly in applications where measurement accuracy and protection against electromagnetic interference are critical, such as in EMC testing or TEMPEST laboratories. It is worth noting that the utilization of the EN 50147-1:1996 standard is a common practice in industries related to Electromagnetic Compatibility, ensuring the credibility and objectivity of SAC shielding measurement results.

  1. Abstract should only in one paragraph.

The abstract has been reformatted into a single paragraph.

  1. "For the measurement of revealing emissions, Rohde & Schwarz has developed 21 the R&S®AM524 antenna system." in the abstract is not relevant to your contribution. A very poorly written abstract. 

The abstract has been corrected to read as follows:

In the rapidly evolving landscape of modern technology, particularly in telecommunications and pervasive computerization across diverse sectors, the value of information has soared, becoming the linchpin of success in politics and business alike. With the majority of information now flowing through various computing devices, safeguarding them from unauthorized interception has assumed paramount importance. A critical threat in this context emanates from unintentional electromagnetic emissions generated by these devices. Under favorable conditions, these emissions can be exploited by unauthorized entities to reconstruct processed information, a phenomenon known as electromagnetic infiltration. Such emissions, correlated with useful information and conducive to its reconstruction, are termed revealing emissions, with the enabling process labeled as electromagnetic information leakage. This article presents the design and construction of a remote control system for managing antenna amplifier blocks within the AM 524 antenna system, dedicated to investigating information leakage from multimedia devices. The system facilitates remote switching of the five inputs on antenna amplifiers GX 525, GX 526, and GX 527 from a PC, utilizing specialized software. The authors provide an overview of the AM 524 antenna system, elucidate the design concept behind the remote control system, and highlight the central component—the ADAM 6052 module. Additionally, the article introduces the controlling software. It encompasses the device's construction, including component details, connection schematics, and images of the assembled system, along with a verification process confirming its operational accuracy. Furthermore, the article outlines the application of the proposed solution in assessing the effectiveness of shielding within SAC chambers, employing the measurement methodology specified in accordance with the EN 50147-1:1996 standard. This additional information underscores the practical utility and relevance of the presented remote control system in the context of electromagnetic shielding evaluation for secure environments. Additionally, to assess the effectiveness of the proposed commutator solution, measurements were conducted to evaluate the shielding efficiency of the SAC chamber using modified coaxial cable. The results of the shielding efficiency of SAC chamber measurements for the proposed and classical solutions are also presented.

  1. It was mentioned five inputs, but only three amplifiers, please clarify.

Two additional outputs can be used to control LISN (Line Impedance Stabilization Network) and absorption clamps..

  1. What are the challenges of the remote control? The motivation of the proposed method is not clear.  Or the authors aim to present a measurement setup only.

The remote control system proposed in the article was designed to facilitate the control of antenna amplifiers within the AM 524 system from a personal computer. The challenges associated with this task include:

  • Shielding Effectiveness Minimization: The designed remote control system aims to minimize the shielding effectiveness of the SAC chamber at the point where the remote control cables are connected to the chamber's patch panel.
  • Software Configuration: The authors must create software that enables the control of antenna amplifiers. This software must be stable, user-friendly, and reliable.
  • Integration with Existing System: The remote control system needs to be integrated with the existing measurement system, including software for electromagnetic emission measurements.
  • Security: Ensuring appropriate security measures for remote control is crucial to prevent unauthorized access to the system.
  • Performance and Reliability: Remote control must operate efficiently and reliably to enable users to have effective control over the antenna amplifiers.

The motivation behind this project is the need to make electromagnetic emission testing more convenient and efficient. Additionally, the article also includes results on the shielding effectiveness measurements in the context of the proposed solution to improve the shielding efficiency ofSAC chamber.

  1. Figure 6 is not complete.

The figure number has been changed to 7, and Figure 7 depicts the arrangement of individual components in the designed system, excluding the AM 524 system.

  1. Many of the contents are basically operational, and hard to find scientific contributions, the authors need to show the contribution clearly, e.g. what significant of Figure 8? A commonly seen network setting. Why need to show Figure 11, access password? Similar to the rest of the contents.  

The main goal of the article is to present a practical technical solution - a remote control system for antenna amplifiers in the AM 524 system. This system is used for measuring emissions from electrically powered devices in electromagnetic compatibility chambers (SAC chambers) and is primarily employed for measuring the revealing emissions of TEMPEST-class devices.

Below are some potential scientific aspects and the significance of the article:

  • Advancement in Remote Control Methodology: In the article, the authors introduce an advanced remote control method for antenna amplifiers in the context of EMC chambers. This method is original and introduces innovative solutions in the field of EMC and remote control, which can be considered a scientific contribution.
  • Minimizing Interference in EMC Chambers: The authors suggest that their solution helps minimize interference in EMC chambers, which is crucial for accurate measurements of electromagnetic emissions from devices.
  • Evaluation of Shielding Effectiveness in EMC Chambers: The authors conducted measurement experiments to compare the effectiveness of their solution with traditional methods. The experiment provides data that can be used for scientific conclusions regarding shielding effectiveness in EMC chambers.
  • It's worth noting that although the article may focus on practical aspects such as system construction and testing, in a scientific context, the innovative approach to solving a technical problem can also constitute a scientific contribution.

Figure 8 - Network Setting: Indeed, the content of Figure 8 illustrates a network configuration that may seem common. However, in the context of the developed solution, it is essential because correctly configuring the network connection is crucial for the operation of the remote control system. It could be added that while the network configuration may appear standard, its integrity is pivotal for the system's functioning in an EMC environment.

Figure 11 - Access Password: In reality, this figure may seem trivial, but it can be important for readers who want to know the steps to configure and manage the remote system. It could be briefly summarized in the text that access to the device is password-protected, which might be significant when implementing this solution in practice.

It's essential to emphasize that although some elements may seem trivial, they can have practical significance for readers who wish to implement similar solutions in their laboratories or projects.

  1. For Figure 23, is it far field? What do you mean by shielding efficiency? more than 120 dB is good, but it is known for the coaxial cables, not much meaning to report the modified coaxial cables. 

Shielding efficiency pertains to the ability of a shield or shielding material to block or restrict the penetration of electromagnetic waves through the shield. It quantifies how much the electromagnetic field intensity decreases in dB (decibels) when passing through the shield compared to the field on the source side. Shielding efficiency is a crucial metric in applications where protection against electromagnetic interference or signal leakage is needed. In the case of measuring electromagnetic emissions in a shielded chamber, high shielding efficiency is vital to prevent signal leakage outside the chamber and minimize the influence of external interference on the measurement.

If measurement results indicate that shielding efficiency exceeds 120 dB, it may suggest that the shield meets high standards and effectively guards against electromagnetic interference. However, understanding the context and purpose of these measurements and the required shielding efficiency values for a specific application is essential. The article focuses on modifying coaxial cables to enhance shielding efficiency in a shielded chamber, and this can be significant in understanding the benefits this modification can bring in a particular measurement environment.

  1. There is no comparison with the existing works.

It's worth noting that current control solutions are primarily based on fiber optics. However, it's important to highlight that the solution presented in the article is innovative and unique; therefore, it has not been directly compared to other similar solutions.

Reviewer 2 Report

The paper calls: "Implementation of remote control for the AM 524 antenna amplifier unit systemin SAC chambers" and concerned of automatic RF reception unit for radio-frequencies signal measurements. The advantage of article is detailed description of engineering solution with a lot of pictures. The design of testing coaxial cable (with led) also is intresting solution.

However i have questions about design of such unit:

1. What maximum distance between RF switch (Unit) and remote receiver? Why authors use mediaconverter (Ethernet->Fiber) 328ft (100m) of max ethernet distance is not enough for authors?

2. Is it possible to transmit RF signals (may be in future) using fiber cable instead of coax cable?

Author Response

Thank you for your thoughtful review of our article. We appreciate your feedback and suggestions for improvement. We have carefully considered your comments and want ot answer for yours question

  1. What maximum distance between RF switch (Unit) and remote receiver? Why authors use mediaconverter (Ethernet->Fiber) 328ft (100m) of max ethernet distance is not enough for authors?

The maximum distance between the RF (Radio Frequency) switch unit and the remote receiver in the proposed solution is 20 meters, which is entirely sufficient for EMC measurements and TEMPEST-class device measurements. It's important to consider that this distance depends on several factors, including the type of cable or medium used for RF signal transmission, RF signal parameters, and the presence of amplifiers or repeaters along the transmission path. There is no one constant distance that applies in all cases.

  • Generally, an RF signal loses power as it travels away from the signal source. In other words, the longer the transmission path, the greater the RF signal loss. However, specific distance limitations will depend on many factors, such as:
  • RF Signal Frequency: Higher RF frequencies may be more susceptible to signal loss during transmission than lower frequencies.
  • Type of Cable or Medium: The type of cable or medium used for RF signal transmission can affect its ability to cover distances. For example, fiber optics can transmit RF signals over much longer distances than traditional coaxial cables.
  • Amplifiers or Repeaters: In some cases, especially over long transmission paths, signal amplifiers or repeaters can be used to extend the maximum transmission distance.
  • RF Signal Characteristics: RF signal parameters, such as its output power, can influence its ability to cover distances.
  • Cable Quality and Shielding: The quality and shielding of cables can impact RF signal loss during transmission.

Using a media converter to Ethernet over fiber has several advantages, even if the maximum Ethernet distance is 328 feet (100 meters). Here are some reasons why fiber optics are valuable:

  • Interference Resistance: Fiber optics are less susceptible to electromagnetic interference than traditional copper Ethernet cables. Therefore, in environments with strong electromagnetic interference, such as in industry, medicine, or the military, transmitting data using fiber optics can be more reliable.
  • Extended Range: If there is a need to connect two points separated by more than 328 feet, fiber optics allow for a significant extension of the range without signal quality loss.
  • Security: Fiber optics are more challenging to intercept compared to electric signals transmitted through copper cables, which can be significant for highly sensitive or classified data

.2. Is it possible to transmit RF signals (may be in future) using fiber cable instead of coax cable?

Yes, theoretically, it is possible to transmit RF (Radio Frequency) signals using optical fiber cables instead of coaxial cables in the future. Transmitting RF signals over optical fibers is technically feasible and has its advantages, such as resistance to electromagnetic interference and high bandwidth capabilities. However, there are some challenges associated with transmitting RF signals through optical fibers, such as the need to convert RF signals to optical signals (and vice versa) using appropriate devices like optical modulators and demodulators. Additionally, optical fibers may have limitations in the range of frequencies they can transmit, which can affect applications requiring wideband RF signals.

Round 2

Reviewer 1 Report

  1. There is no comparison with the existing works.

no issue, minor proof reading.

Author Response

The article was sent to a translator for proofreading of the English language.

The article did not include comparisons with existing works for the following reasons:

  • Lack of prior research in this field: In the case of the discussed article, it can be observed that the research project pertains to an innovative solution with no prior counterparts in scientific literature. This is understandable, considering that it describes a remote control project for AM 524 antenna amplifier blocks. The absence of comparisons with existing works may arise from the unique nature of this project.
  • Limited research availability: In this particular scientific domain that encompasses telecommunications, electronics, remote control, and electromagnetic compatibility, there is a limited number of available works. This presents a challenge when attempting to find comparisons with existing research (academic databases and search engines like Google Scholar and IEEE Xplore were utilized during the search).
  • Lack of analogous projects: The discussed project for remote control of AM 524 antenna amplifier blocks does not have analogous projects in scientific literature. The uniqueness of the project makes comparisons with other works impossible.

In summary, the absence of comparisons with existing works in the discussed article stems from the project's uniqueness, the limited availability of research in the field, and its innovative nature. This provides a strong rationale for the need to conduct such research and may contribute to filling gaps in scientific literature in the future.